# Experimental Study and Numerical Simulation Analysis on Vertical Vibration Performances of 12 m Span Wood Truss Joist Floors

Zhanyi Zhang [1], Shuangyong Wang [2], Hao Deng [2] and Haibin Zhou [1,2,*]

1   Research Institute of Wood Industry, Chinese Academy of Forestry, Beijing 100091, China
2   National Engineering Research Center of Wood Industry, Beijing 102300, China
*   Correspondence: zhouhb@caf.ac.cn

**Abstract:** Walking-induced vibration control in wood floors is a critical issue attracting the attention of many researchers and engineers. This paper presents an experimental study applying static deflection tests, modal tests, and pedestrian load tests to a series of full-scale 12 m span tooth plates connected to wood truss joist floors with strongbacks and partition walls. A comparison of the calculation error of vibration parameters between the theoretical formula and a numerical model was also conducted. The results show that strongbacks and partition walls effectively reduce both the vertical displacement and the root means acceleration at the center of the floor under pedestrian load but increases the natural frequency. The partition wall can achieve a better vibration-reduction effect than strongbacks. The error of the finite element model is higher than that of the theoretical formula. Using the theoretical formula in engineering wood floor design is recommended.

**Keywords:** wood floor; floor vibration; vibration serviceability; numerical methods





## 1. Introduction

A raised wood floor system is generally designed in residential low-rise construction to elevate the living space off the ground or downstairs with the benefit of a high degree of industrial prefabrication. The vertical vibration performance of wooden floors is essential to residential building quality. However, raised wood floors are sensitive to residents' daily activity or other dynamic loads, and annoying vibrations arise from their low mass compared with steel or concrete floors. Research shows that if the natural vibration frequency of the floor is between 4 and 8 Hz, residents will feel discomfort and anxiety due to the similar resonances of the wood floor with organs of people, which affects the comfort and livability of wooden buildings [1]. Currently, the serviceability design of wood-framed floors is usually based on limiting the relevant parameters such as deflection, acceleration, natural frequency, or their combinations [2]. This approach is practical for vibration control of small and medium span floors; whether it suits large span floors remains unclear. Therefore, it is necessary to analyze the vertical vibration performances of large-span wooden floors through relevant experimental studies and summarize a general design method for the serviceability of wooden floors.

It is necessary to evaluate wood floor theoretical models with sufficient field test data to study the dynamic performance of wood floors. Previous research studies related to full-scale tests are as follows. Khokhar et al. [3] conducted experimental tests on 4.2 m laminated veneer lumber (LVL) joist wood floors and compared different types of between-joists bracing on the effects of vibrational serviceability. Jarnerö et al. [4] assessed the dynamic performances of 5.1 m wood floors experimentally in the laboratory with different boundary conditions and in field tests at different stages of construction. Weckendorf et al. [5] presented an experimental study of low amplitude dynamic responses on 5.5 m cross-laminated-timber (CLT) floors. Ding et al. [6] conducted vibration tests on 6 m spruce-

pine-fir (SPF) timber joist floors. Wang [7] investigated the structural behavior of 6.1 m two-way wood truss floors. Zhou et al. [8,9] analyzed the vibration performances of 4.7 m solid lumber joist floors and 8.26 m engineered wood truss joist floors. Xue [10] presented an experimental study on 6 m wooden truss joist floors. Rajendra Rijal et al. [11] compared the modal behaviors of 6 m and 8 m timber floors. Studying other factors related to the vibration performance of wood floors is also necessary. Onysko et al. [12] conducted massive vibrational serviceability tests on floors with a span of less than 10 m. Foy Cdric et al. [13] conducted modal tests on two 4 m wood floors in free boundary conditions and built a numerical model to carry out the parametric study. Fuentes et al. [14] presented an experimental study of a 7.2 m wood floor. Xue et al. [15] studied the effects of joist spacing and bracing elements on 6 m wood truss joist floors. Persson et al. [16] analyzed the influence of uncertain parameters on the modal properties of 7.2 m plywood truss floors. Yujian Dong and Lilin Cao. [17] proposed a model to determine the human-induced response of a 9 m steel-wood composite floor. Zhang and Yang [18] compared loading methods on floor vibration due to individual walking styles. Sepideh Ashtari [19] analyzed the difference between the rigid and flexible connections of a 10.8 m CLT floor.

The classical measurement approach using a modal hammer in the case of experimental modal analysis is time-consuming and laborious. Some modern and contactless methods have been developed. LukaszScislo [20] applied a 3D scanning vibrometry system, a non-contact measurement method, to obtain natural frequencies and modal shapes of ultra-light structures. The results show that this system is helpful for modal analysis of high fragility and low weight structures without contact by using the excitation of the loudspeaker. Emilio Di Lorenzo et al. [21] investigated the use of digital image correlation (DIC) for modal analysis. DIC is a non-contact full-field image analysis technique that uses high-speed and high-resolution cameras to measure structures' strains and displacements to derive the structure's modal characteristics.

Vibration serviceability research of wood floors is usually concerned with trusses ranging from 3 m to 12 m. However, the static and dynamic performances related to the vibration serviceability of wood truss joist floors longer than 9 m have not been clarified to date. This paper analyzes the vertical vibration performances of 12 m wood truss floors by field tests. Tests at this scale are rare in related research. It also discusses the effects of strong-backs and partition walls on vibration responses. Laboratory studies based on numerical simulations are used to improve our understanding of the complexities of the vibration response of large span raised wood floor systems. A 12 m finite element model of a wood floor is built to predict modal behaviors and unit point load deflection. The simulation results are compared with theoretically predicted results to evaluate the finite model of wood floors. This paper intends to contribute to understanding the vibration performances of large-span wood floors for future vibration serviceability research and the engineering application of large-span wood truss joist floors.

## 2. Overview of the Wood Floor and Test Methods

### 2.1. Floor Configurations

The floor was designed according to the standard of the Canadian National Building Code with a deflection of not more than L/360 under a uniform load of 1.9 kPa. Based on the edge of the surrounding wall, the design length of the floor was 12.11 m, the design width was 6.09 m, and it was built on a wall with a height of 1.85 m, as shown in Figure 1. The wall frame material was SPF material, covered with OSB board, and the walls were assembled with 50 mm nails at intervals of 300 mm. The wooden truss joists consisted of J-level SPF material, the section size was 38 mm × 89 mm, and the top and bottom chords were connected by SPF finger joints and glued in the thickness direction. Tooth plates connected the nodes of the truss. The dimensions of the truss and the tooth plates are shown in Figure 2. The 21 trusses were arranged on the wall with a spacing of 300 mm. The ends of each truss were nailed obliquely with the top plate of the wall by two 125 mm drill-tail screws. The rim boards were made of LVL, 38 mm thick and 500 mm high. The

rim board and the truss were vertically nailed with three 70 mm drill-tail screws. The rim boards and the outermost wooden truss joists were placed on the wall, which was obliquely nailed symmetrically on both sides at intervals of 600 mm with the top plate of the wall by 90 mm self-tapping screws. The details of nail connections are shown in Figure 3. The sheathing material was 15 mm OSB board, 2440 mm long and 1220 mm wide. The major direction of the OSB sheathing was placed perpendicular to the joist. The OSB subfloor was arranged half-staggered from each other, as shown in Figure 4. The OSB subfloor was connected to the joist using 50 mm screws of 150 mm around the perimeter and 300 mm in the field. The distance between the screws and the OSB subfloor edge was greater than 10 mm.

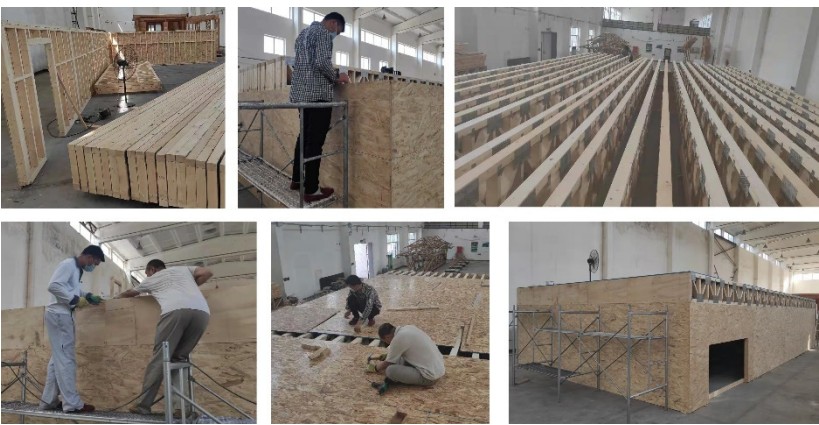

**Figure 1.** Construction of test floor.

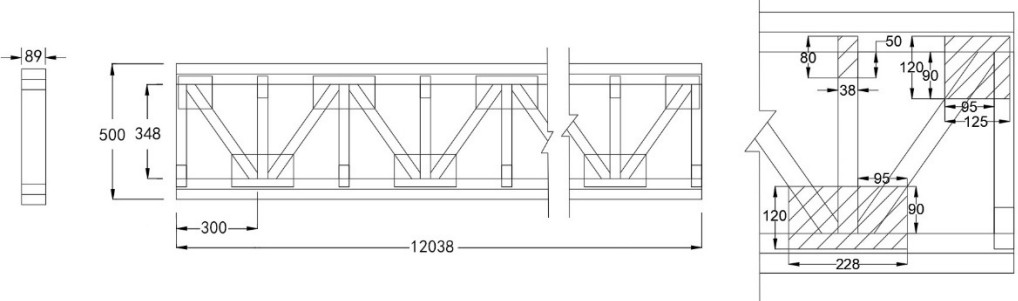

**Figure 2.** Dimensions of timber truss (unit: mm).

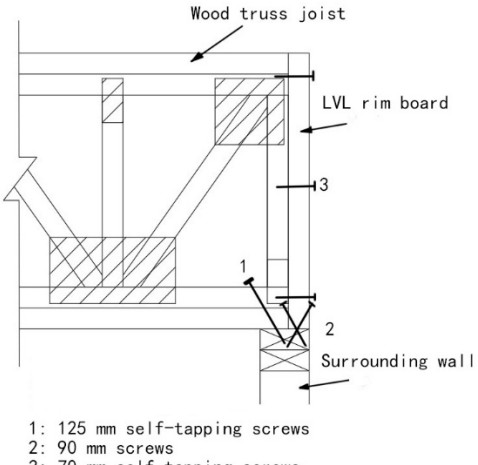

1: 125 mm self-tapping screws
2: 90 mm screws
3: 70 mm self-tapping screws

**Figure 3.** Connection between the floor and the wall.

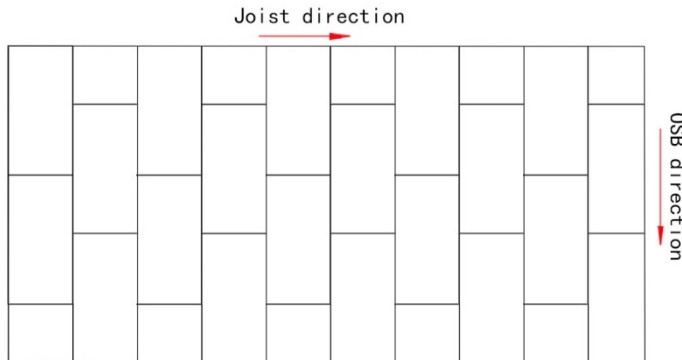

**Figure 4.** OSB layout.

Different test floors were designed by installing strongbacks or partition walls into the initial floor system. The information for each group of test floors is shown in Table 1, and the schematic diagram is shown in Figure 5. The strongbacks were constructed from 40 mm × 140 mm cross-sectional size, J-level SPF specification material selected by visual inspection. It is installed in the vertical direction of the joists and connected to the truss web rod with three 80 mm wood screws from the top, middle, and bottom in the height direction of the strongbacks, as shown in Figure 6. The door on the surrounding wall could transport the partition wall. It is connected with the interface at the corresponding position on the surrounding walls, as shown in Figure 7.

**Table 1.** Details of test floor structures.

| Floor | Details |
|---|---|
| T1 | No strongbacks and partition wall |
| T2 | Double strongback rows at mid-span |
| T3 | Double strongback rows at mid-span and one strongback row each at one-sixth span and one-third span |
| T4 | One partition wall each at one-fourth span |
| T5 | One partition wall at mid-span |

**Figure 5.** Schematic representation of test floors.

The studs of the partition wall and the surrounding wall were fastened with two 90 mm drill screws at intervals of 600 mm, and the top and bottom plates of the partition wall and the surrounding wall were fastened with two 50 mm drill screws.

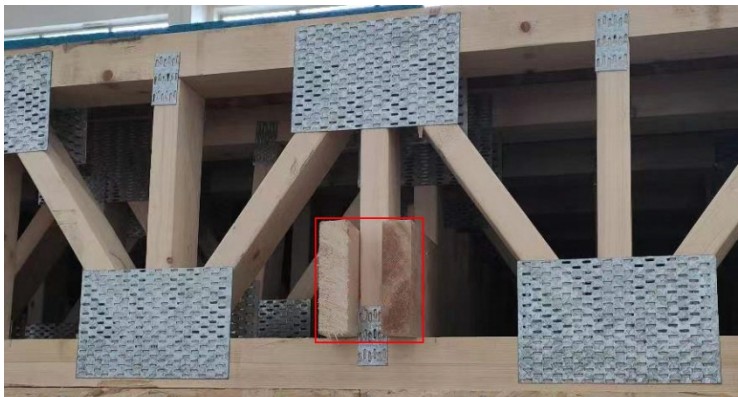

**Figure 6.** Double strongbacks rows at mid-span.

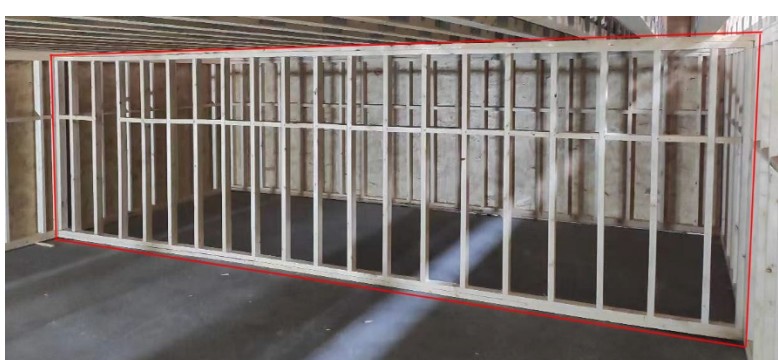

**Figure 7.** Partition wall.

*2.2. Test and Analysis Methods*

2.2.1. Test Methods

1. Static load deflection test

The static load deflection of 1 kN is a common parameter in the vibration design of wooden floors. The test apparatus included a Mitutoyo brand ID-C150XB dial indicator with an accuracy of 0.001 mm and a range of 0 to 50.8 mm, seven 180 cm steel hangers, and 1 kN weights consisting of five 20 kg iron discs. The layout of the test measuring points is shown in Figure 8. Except for T5, which was measured at the quarter-span marked as line B, the rest of the floors were measured at the mid-span marked as line A. The measuring point was the intersection of the measuring line and the joist, and the loading point coincides with the measuring point on the J10 joist.

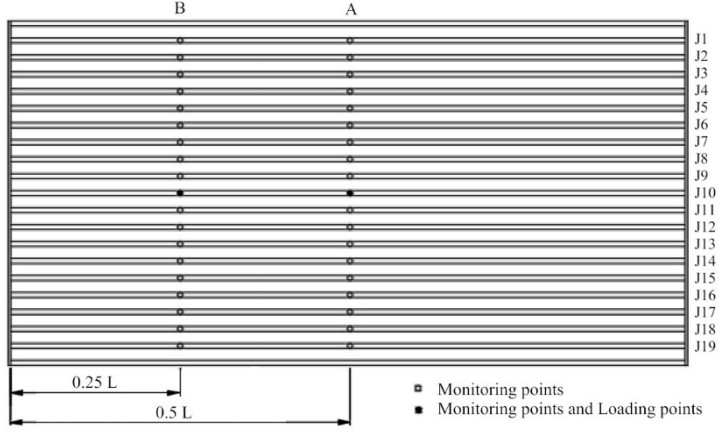

**Figure 8.** Distribution of the monitoring points and loading points.

There were 21 joists on the test floor, except for the two joists nailed to the wall on both sides. There was a total of 19 monitoring points A maximum of 7 points could be measured simultaneously, so the measurements were performed three times. The measurement process is shown in Figure 9. First, seven steel hangers were placed under the monitoring point, and dial indicators were fixed on each steel hanger, ensuring that the dial indicator probe was in contact with the monitoring point marked on the lower surface of the joist. The zero-adjustment operation was then performed. After that, the test personnel moved the heavy object to the loading point and left the floor. The display data was recorded when it was stable and summarized as the static load deflection curve of the 1 kN concentrated load at each joist measuring point of the floor.

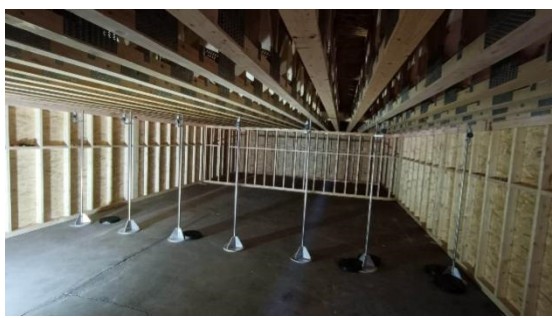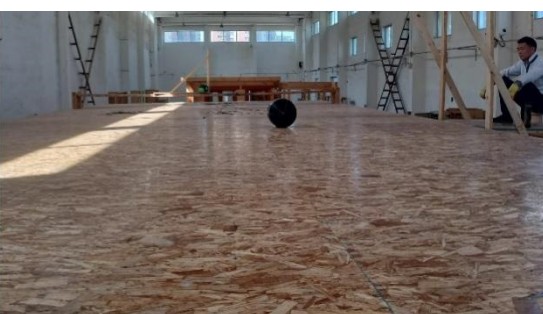

**Figure 9.** 1 kN static load deflection test.

2. Modal test

The modal test apparatus included the INV9314 test hammer (COINV Inc., Beijing, China) with a reference sensitivity of 50.5 uV/N, the INV3020C signal acquisition and analyzer with the DASP-V10 software platform (COINV Inc., Beijing, China) that can collect and analyze vibration signals, and four INV9828 piezoelectric acceleration sensors (COINV Inc., Beijing, China) with a sensitivity of 50 mV·s$^2$/m, and a range of 100 m/s$^2$.

A total of $20 \times 6 = 120$ monitoring points were marked on the floor, as shown in Figure 10. The bottom of the four sensors was then coated with beeswax to ensure they could connect closely with the floor surface and placed on measuring points 1–4. An excitation point was at the center of the building, and if the excitation effect was not clear, it was selected at the quarter span. A rubber head hammer was used to excite the floor three times to obtain the acceleration response under the excitation action. The sensor was then moved to measuring points 5–8, the floor was excited three times, and the process was repeated until all the points were measured. The test personnel pounded the floor on the prefabricated wooden beams to ensure that there was no additional mass on the floor, as shown in Figure 11. The collected acceleration responses were processed by fast Fourier transform (FFT) to obtain the frequency response function (FRF), and then the first three order natural frequencies, damping, and mode shapes of the floor were extracted from the FRF.

3. Pedestrian load test

The pedestrian load test apparatus was the same as the modal test. The walking paths for each group of floors are shown in Figure 12. The testers weighed 85 kg, and a metronome was used to adjust the walking frequency to around 2 Hz. By walking along three paths, horizontal (H), vertical (S), and oblique (X), walking excitation was applied to the floor. In order to prevent the test personnel from accidentally touching the sensor while walking, the walking path was set at 60 cm wide. The sensor was placed at point A on the central joist of the floor. When the tester started to walk, the data was recorded by the signal acquisition analyzer, and the recording stopped when the tester reached the end of the path, as shown in Figure 13. From the pedestrian load test, the time-history curve of the acceleration response of the floor under walking load was obtained, and the root-mean-square (RMS) acceleration at the center of the floor was determined.

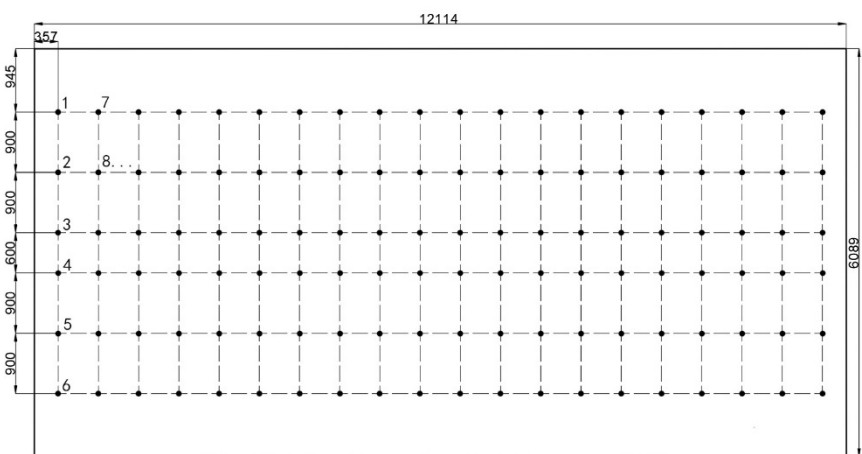

**Figure 10.** Distribution of the monitoring points (unit: mm).

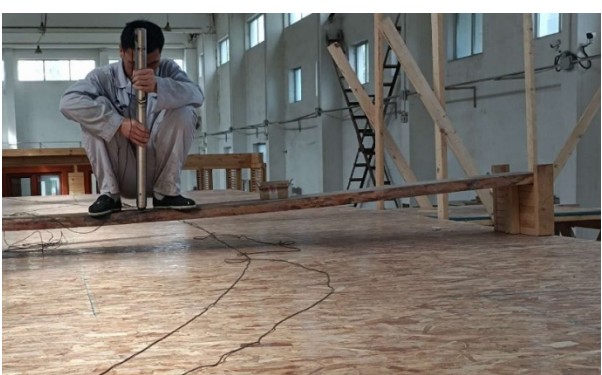

**Figure 11.** The modal test.

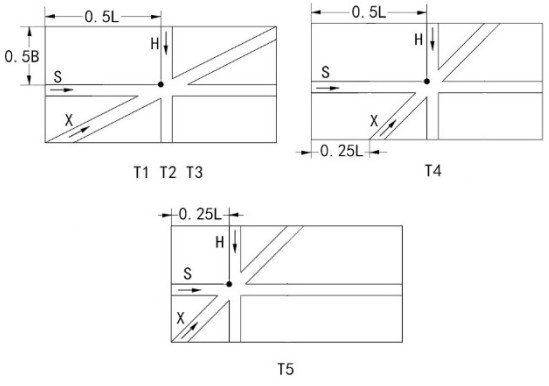

**Figure 12.** Layout plan of pedestrian load test.

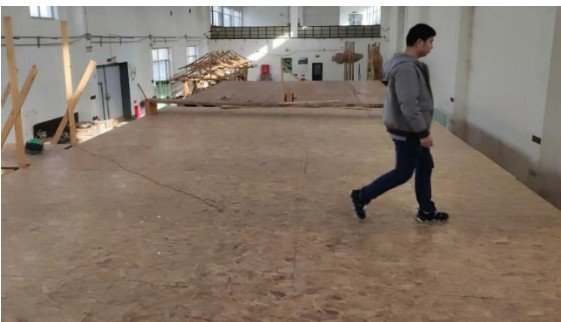

**Figure 13.** Pedestrian load test.

### 2.2.2. Analysis Methods

1.     Finite element method

The finite element model of the 12 m wood truss floor consists of the truss and the sheathing. Each truss member adopts the Beam 188 element, and the sheathing adopts the Shell 181 element. The material parameters of the SPF and OSB sheathing are shown in Table 2. All the nodes are set as hinges. The common boundary conditions of the floor include simply supported on four sides, fixed on four sides, and simply supported on both sides and fixed on both sides. The nail connection between the ends of the truss and the wall can be seen as a half-stiffness node. So, it is used in the numerical model that the fixed support on the edge joists and the simply supported connection between the other joists. 1 kN concentrated load is applied at the center of the floor.

**Table 2.** Parameters of the engineered wood materials.

| Properties | SPF | OSB |
| --- | --- | --- |
| | **38 mm × 89 mm** | **15 mm Thickness** |
| $\rho$ (kg/m$^3$) | 497 | 650 |
| $\nu$ | 0.49 | 0.45 |
| $E_L$ (MPa) | 8700 | 4280 |
| $E_R$ (MPa) | 900 | 2080 |
| $E_T$ (MPa) | 700 | 20.8 |
| $G_{LR}$ (MPa) | 500 | 1000 |
| $G_{LT}$ (MPa) | 500 | 50 |
| $G_{RT}$ (MPa) | 30 | 50 |

2.     Theoretical method

The test floor is simplified as in Figure 14. Based on the Timoshenko ribbed plate theory, the first-order vibration frequency of the wooden floor and the deflection of the wooden floor under the concentrated load at the center of the slab can be calculated. This paper adopts the first-order natural frequency calculation formula from [22] and the deflection calculation formula from [23].

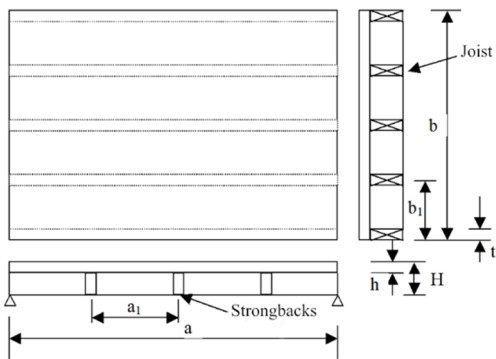

**Figure 14.** Timber floor ribbed slab model.

## 3. Results and Discussion

### 3.1. Influence of Strongbacks and Partition Wall on the Vertical Deflection of the Floor

The 1 kN concentrated load deflection of each group of floors is shown in Figure 15. Among them, the 1 kN static load deflection of T1 was the largest (1.376 mm), which is less than L/250, and meets the bending deflection limit for floor beams according to GB 50005-2017 "Standards for Design of Timber Structures" [24]. The deflection of the T2 floor was 39.3% lower than that of the T1 floor. It shows that the strongbacks can significantly increase the stiffness of the floor. GB 50005-2017 [24] also stipulates that the spacing of the strongbacks in the span direction should not be greater than 2.1 m. It can

be seen from Figure 15 that the static load deflection of T3, which meets this condition, is further reduced compared with T2 after installing two rows of strongbacks. Compared with T1, the deflection of T3 decreased by 47.5%. For the long-span wooden floor, when the distance between the strongbacks was not more than 2.1 m, the rigidity of the floor was significantly improved compared to when the strongbacks were not installed. Compared with T1, the deflection of T4 decreased by 61.3%, indicating that installing a partition wall can significantly increase the stiffness of the floor. Compared with T1, the deflection of T5 decreased by 64.4%, which was similar to T4. It may be related to the space with a span of 6 m divided by the partition wall.

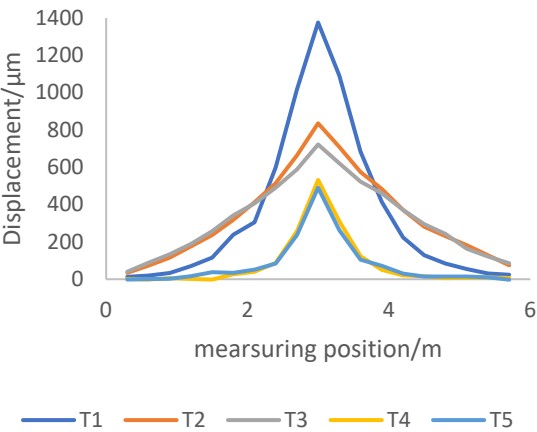

**Figure 15.** The measured deflection of test floors.

### 3.2. Influence of Strongbacks and Partition Walls on Modal Performances of Floors

As shown in Figure 16, the first three vibration modes of the five groups of floors were similar, up and down in the form of sine waves along the width direction. The first three natural frequencies and damping of the five groups of floors are shown in Table 3. The first-order vibration frequency of T1 is 6.8 Hz, less than 8 Hz, which is the limit to vibration comfort. Compared with T1, the first-order frequencies of T2 and T3 increased by 3.0% and 6.9%, respectively, the second-order frequencies increased by 21.6% and 36.0%, respectively, and the third-order frequencies increased by 55.2% and 80.3%, respectively. These results indicate that the installation of strongbacks improves the first three frequencies of the floor, and the effect on the second and third frequencies is noticeable. The test results are consistent with the installation of strongbacks for 6 m-span wood truss joist floors [10]. It was demonstrated that the first-order frequency of the floor is mainly controlled by the stiffness of the floor parallel to the joist direction. After installing the partition wall, the first three natural frequencies of the floor were significantly improved, and the spacing of adjacent order frequencies was also improved. It indicates that partition wall increases the stiffness of the floor and improves vibration comfort. Compared with T1, the first-order frequency of T5 was increased by 77.6%, which indicates that the installation position of the partition wall has a different effect on the stiffness of the floor, and the partition wall in the middle has the best effect on the natural vibration frequency. Compared with T4, the first-order frequency of T5 was reduced by 2.5%. The maximum span of T4 and T5 divided by the partition wall was 6 m. At the same time, the first-order frequency of T4 and T5 was close, indicating that the maximum distance between the walls perpendicular to the joist direction affects the first-order frequency of the floor. Compared with T4, the second and third order frequencies of T5 increased by more than 10%, indicating that the increase of longitudinal stiffness by installing a partition wall at a 6 m span is better than at 3 m and 9 m spans.

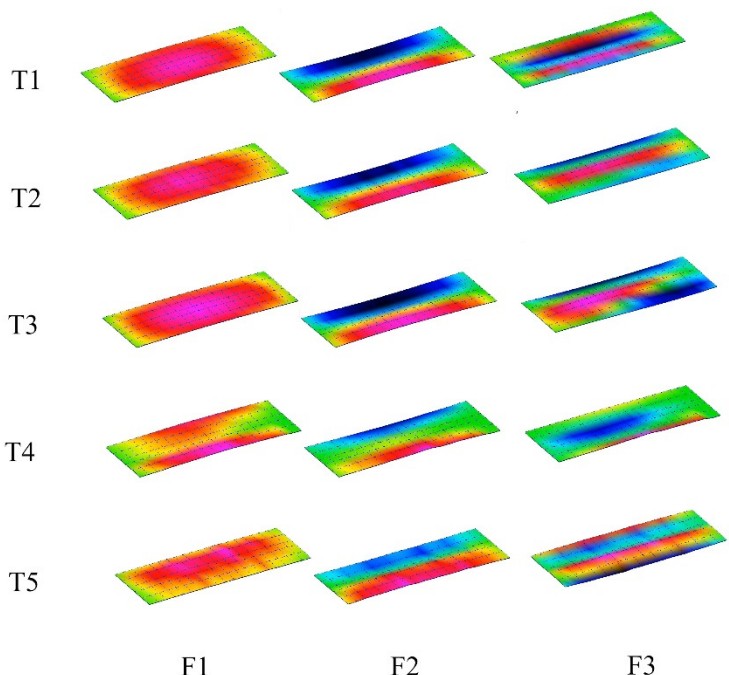

**Figure 16.** First three mode shapes obtained from the experiment.

**Table 3.** The first three-order natural vibration frequencies and damping of each test floor.

| Floor | $f_1$ (Hz) | $\zeta_1$ (%) | $f_2$ (Hz) | $\zeta_2$ (%) | $f_3$ (Hz) | $\zeta_3$ (%) |
|-------|-----------|---------------|-----------|---------------|-----------|---------------|
| T1 | 6.8 | 1.3 | 8.1 | 0.8 | 9.1 | 1.3 |
| T2 | 7.1 | 1.4 | 9.9 | 1.4 | 14.1 | 1.9 |
| T3 | 7.3 | 0.9 | 11.1 | 2.0 | 16.4 | 1.4 |
| T4 | 12.5 | 3.5 | 15.1 | 2.4 | 16.8 | 1.9 |
| T5 | 12.2 | 3.4 | 16.8 | 3.5 | 19.4 | 1.7 |

*3.3. Influence of Strongbacks and Partition Wall on Pedestrian Load Response of Floor*

The pedestrian load consists of a series of single footfall loads. The time history curve of floor vibration response is obtained by recording the vibration response caused by pulse excitation under a continuous walking load. The time history curve of vibration acceleration response at the center of the floor is shown in Figures 17–21 for each floor along three walk paths. When the natural frequency of the floor was greater than 8–10 Hz, the floor produced transient vibration and decayed rapidly. The amplitude of transient vibration is related to the stiffness and quality of the floor. When the natural frequency of the floor is less than 8–10 Hz, the floor may produce resonance, and the amplitude is related to damping. From the time history curve, it can be found that T1, T2, and T3 floors with natural frequencies less than 10 Hz had obvious resonance under walking load. The acceleration response raised by the pedestrian load remained at a high level even at the beginning of walking, then slowly reduced after the person stopped walking. The T4 and T5 floors with natural frequencies greater than 10 Hz generated transient vibration under walking load and decayed rapidly. The acceleration response was high only when the foot fell, and the peak acceleration increased fast when walking past the sensor.

The RMS acceleration of central vibration of all floors along three walking paths is shown in Table 4. It shows that the RMS vibration acceleration at the center of each floor was greater than that in the H direction when walking in the S direction. When walking in the three paths, the RMS vibration acceleration at the center of the T1, T2 and, T3 floors was about twice that of T4 and T5 floors. The main reason is that the natural vibration frequencies of these three groups of floors were low, and the floors were resonant under walking excitation. Compared with the pedestrian load test results of the same type of

6 m-span timber floor, the RMS acceleration of a 12 m-span floors in all directions was reduced by more than 60% [10]. However, the level of RMS acceleration of all wood floors is too large to satisfy the vibration serviceability Standard in ISO 10137 [25].

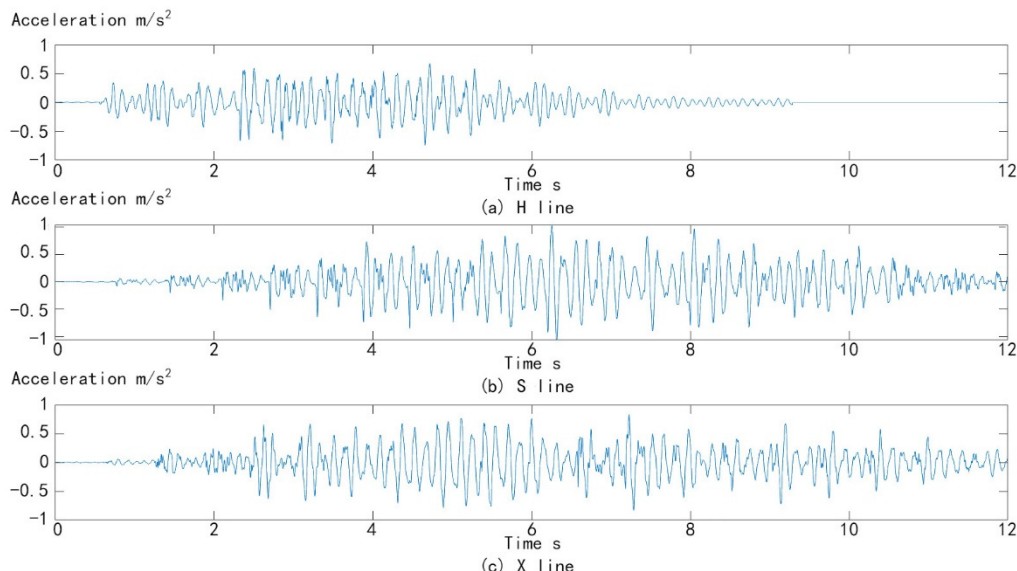

**Figure 17.** Acceleration response of T1 floor.

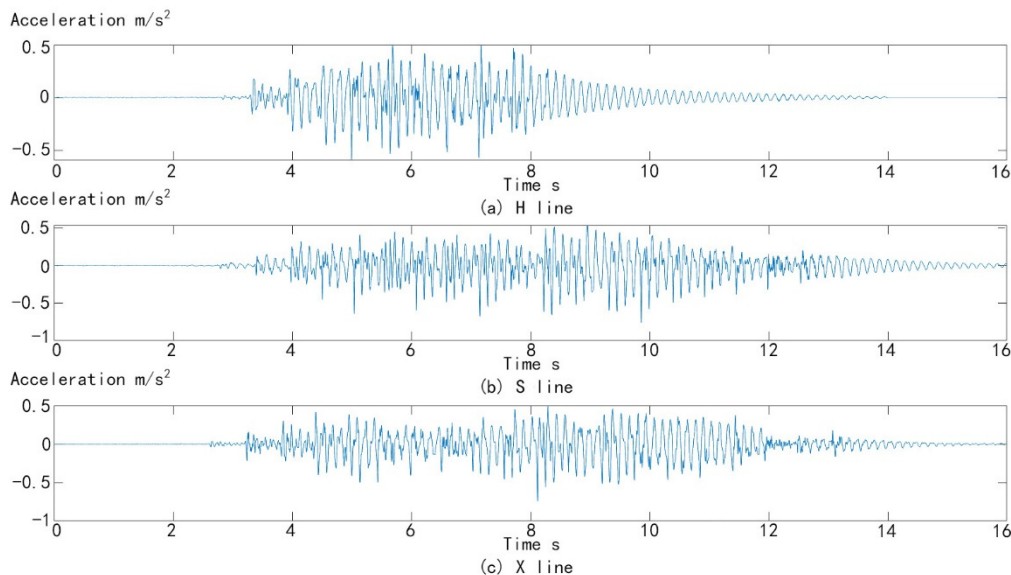

**Figure 18.** Acceleration response of T2 floor.

**Table 4.** RMS acceleration of the floor under a single person walking load (unit: m/s$^2$).

| Floor | H Line | S Line | X Line |
|:---:|:---:|:---:|:---:|
| T1 | 0.176 | 0.255 | 0.197 |
| T2 | 0.111 | 0.131 | 0.123 |
| T3 | 0.127 | 0.170 | 0.157 |
| T4 | 0.051 | 0.070 | 0.050 |
| T5 | 0.045 | 0.064 | 0.051 |

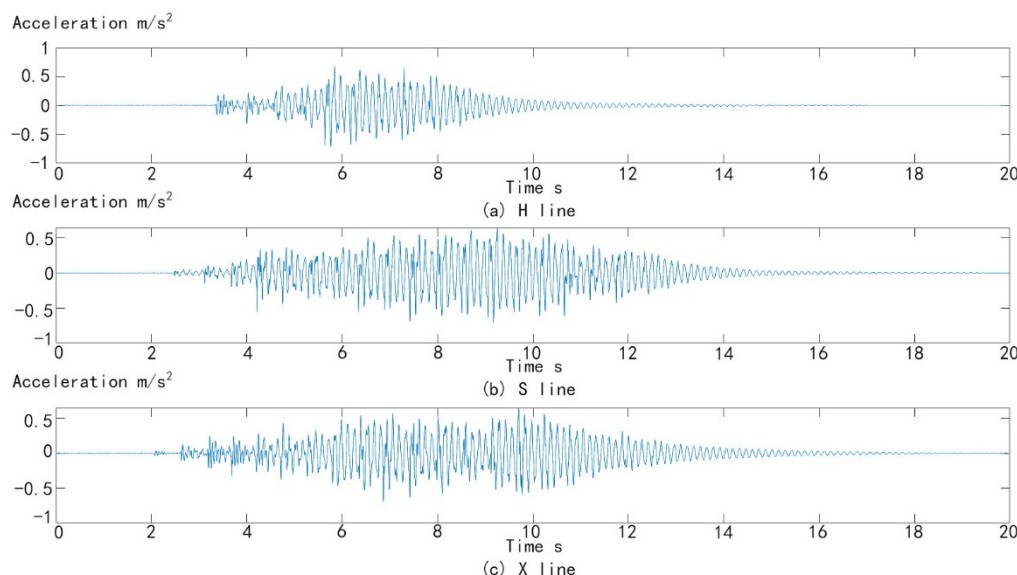

**Figure 19.** Acceleration response of T3 floor.

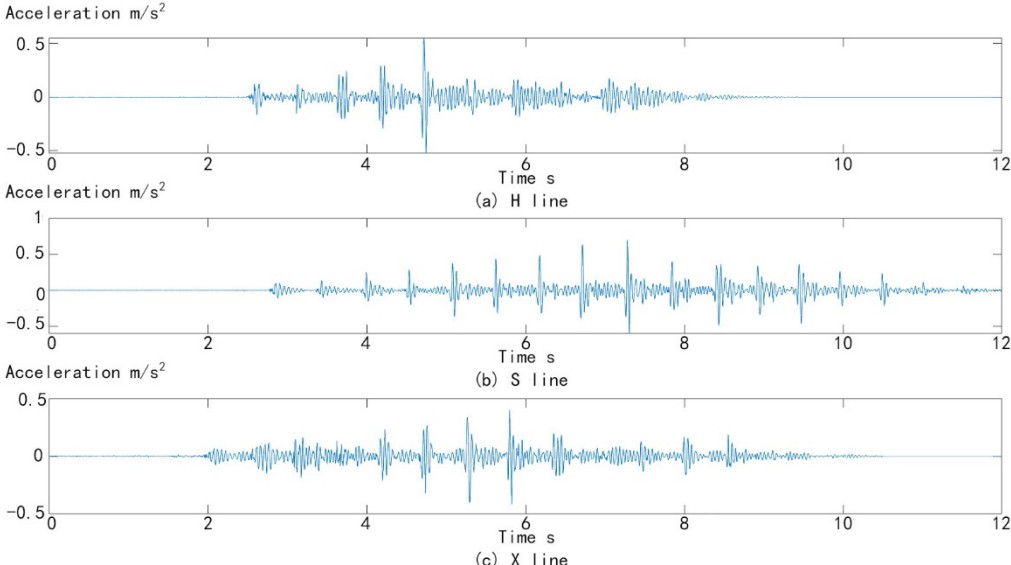

**Figure 20.** Acceleration response of T4 floor.

After two rows of strongbacks were installed in the center of the floor span, the RMS vibration acceleration at the center of the floor decreased when walking along the three paths. It was related to the increase in floor stiffness after the installation of strongbacks. When the strongbacks were installed two meters from the center of the floor span, the RMS vibration acceleration at the center of the floor did not decrease but increased. In summary, strongbacks in the center of the floor span reduce the RMS vibration acceleration at the center of the floor. It decreases the transient vibration response of the floor caused by the walking load. For floors with lower natural frequencies, the priority is to increase the natural frequency and then find other ways to reduce the vibration response.

After the installation of the partition wall, the RMS acceleration of vertical vibration of the floor under walking load was significantly reduced. Compared with T1, the RMS vibration acceleration at the center of the T4 floor decreased by 70.8% when walking in the H direction, 72.4% when walking in the S direction, and 74.8% when walking in the X direction. Compared with T1, the RMS vibration acceleration at the center of the T5 floor decreased by 74.4% when walking in the H direction, 75% when walking in the S direction,

and 74.3% when walking in the X direction. The different positions of the partition wall had different effects on the improvement of the fundamental natural frequency of the floor, so the reduction of the RMS acceleration was also different. By comparison, installing the partition wall in the center of the floor span was more effective at reducing the RMS vibration acceleration in the center of the floor. The natural frequencies of T4 and T5 were similar, and there was little difference in the RMS vibration acceleration in the center of the floor when walking in various directions. It shows that the RMS vibration acceleration at the center of the floor may be related to the maximum distance between the walls perpendicular to the joist direction. The larger the distance, the smaller the RMS vibration acceleration at the center of the floor.

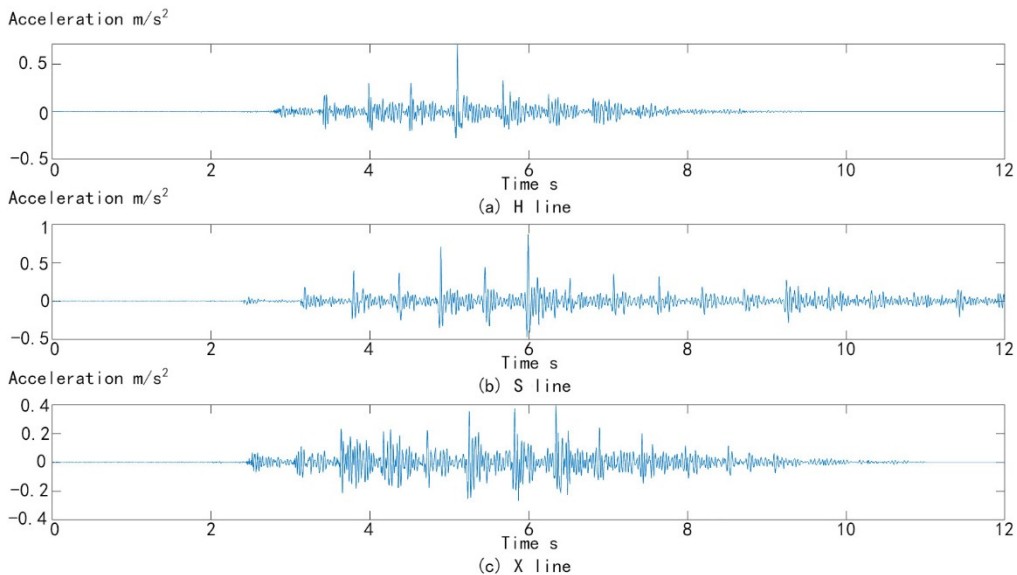

**Figure 21.** Acceleration response of T5 floor.

*3.4. A Comparison of the Numerical Simulation and Theoretical Results*

The predicted value of T1 deflection can be obtained by the finite element software static analysis and theoretical formula solution. The comparison between the two predicted values and the measured value is shown in Table 5. It shows that the theoretical prediction value of 1.59 mm is 15% higher than the measured value of 1.376 mm, which may be attributed to the calculation formula being different from the actual floor. The boundary conditions of the theoretical model assume simply supported on four sides, whereas the outermost joist, in actuality, is nailed to the surrounding wall. Therefore, the boundary conditions of the existing wood floor system are complex. The tooth plate joints of the actual wood truss joist are semi-rigid connections, so the overall stiffness of the floor is higher than theoretically predicted, which leads to the theoretically predicted deflection being large.

**Table 5.** Comparison between the measured and predicted deflection values under 1kN static load at the center of the T1 floor.

| Predicted Values (mm) | | Experimental Value (mm) |
|---|---|---|
| **Theoretical Predicted Value** | **Simulation Value** | |
| 1.59 | 1.79 | 1.376 |

3.4.1. Simulation of 1 kN Static Load

The simulation results of the deflection of the T1 floor under the concentrated load of 1 kN acting on the floor center are shown in Figure 22. The deflection at the center of the floor is the largest, and the calculated result is 1.79 mm, which is 30% higher than the

measured value. The error is larger than the experimental value, which is similar to the floor simulation results of Shen [26]. The deflection distribution is oval. Parallel to the span direction, the deflection decays slowly faster when vertical to the span direction. The simulation predicted value of the floor is greater than the theoretical predicted value, and there is a substantial difference from the measured value. Therefore, the model could be further optimized by increasing the rim board model and refining the truss joints from hinged joints to semi-rigid connections.

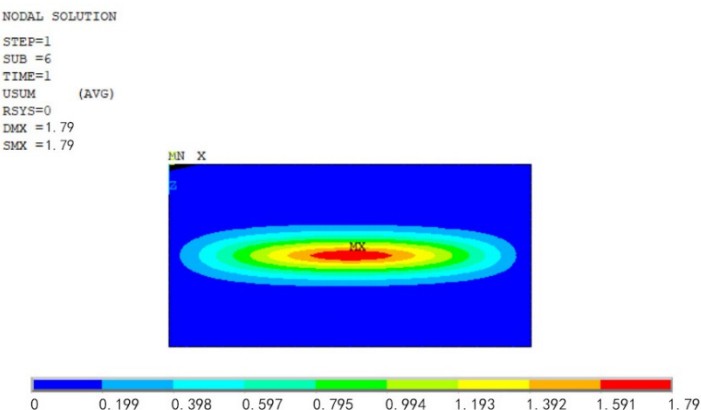

**Figure 22.** Vertical displacement distribution of timber floor model (unit: mm).

### 3.4.2. Modal Test Simulation

The first three vibration modes of the finite element floor are shown in Figure 23, and the first three natural frequencies are 5.88 Hz, 6.03 Hz and 6.37 Hz. The theoretical predicted value of the first-order natural frequency of the floor can be calculated. A comparison of the predicted and measured values of the first-order natural frequency of the floor is shown in Table 3. It can be seen from Table 6 that the predicted values obtained by the two methods are lower than the measured values. There is little difference between the theoretical predicted values and the measured values, indicating that it is feasible to estimate the first-order vibration frequency of the floor by using the theoretical formula. The first three modes of the simulated floor coincide with the measured floor. The difference between the predicted value and the measured value of the first-order frequency is 13.5%, which is close to the error of the finite element model of the 6 m wooden truss joist floor established by Shen [26]. However, the error of the second-order and third-order natural frequency values is more than 20%. Therefore, when the boundary conditions are the fixed support on the edge joists and the simply supported connection between the other joists, the modal performance of the floor can be predicted. However, the accuracy of the first three natural frequencies is still unacceptable.

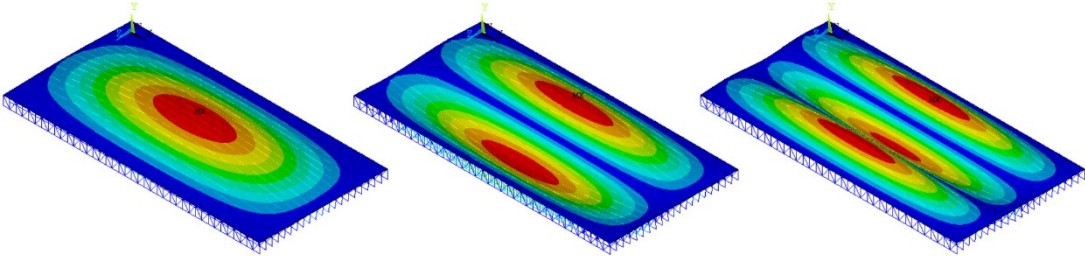

**Figure 23.** The first three modes of the numerical model of the floor.

**Table 6.** Comparison between the measured and predicted values of the first-order natural frequency of wood floors.

| Predictive Value (Hz) | | Experimental Value (Hz) |
|---|---|---|
| Theoretical Predictive Value | Simulation Value | |
| 6.47 | 5.88 | 6.8 |

## 4. Conclusions

In this paper, laboratory tests including static load deflection tests, modal tests, and pedestrian load tests were conducted to reveal the vibration performances of 12 m wood truss joist floors. The selective action law of strongbacks and partition walls on the vibration performance of large span wood floors is discussed. Combined with numerical simulation, the prediction accuracies of the finite element model and theoretical formula on the static load deflection and modal performance of the floor are compared.

The strongbacks increase the stiffness of the floor perpendicular to the joist direction, which can significantly increase the high-order natural frequency of the floor, but has little effect on the fundamental natural frequency. Installation of two rows of strongbacks in the midspan can significantly reduce the RMS vibration acceleration at the center of the floor under walking excitation. However, the RMS acceleration increases with increasing numbers of strongbacks. Installation of a partition wall perpendicular to the joist under the floor can improve the overall stiffness of the floor, and the effect on the fundamental natural vibration frequency and RMS acceleration of the floor is better than that of the strongbacks.

The 1 kN static load deflection of the 12 m span wood floor is 1. 376 mm. The prediction error of the theoretical calculation formula based on the ribbed plate model of the wooden floor is 15.6%, and the error of the finite element model is 30.1%. The fundamental natural vibration frequency of the 12 m span floor is 6.8 Hz, the error of theoretical calculation is 4.9%, and the error of the finite element model is 13.5%. In engineering applications, The vibration performances can be estimated by the theoretical formula and qualitatively analyzed by a finite element model.

This study presents a method to build a large-scale test platform of wood floors. This assessment of the vibration performances of wood floors will benefit the engineering application of large-span wood truss joist floors. Further research will be conducted on other methods to increase the stiffness of wood floors and the effect of different loads, such as multi-person walking loads, on the vibration performance of large span wood truss joist floors. A more accurate model also will be developed to predict the vibration responses of wood floors.

**Author Contributions:** Z.Z.: conceptualization, field measurements, numerical modelling, original draft preparation; S.W., H.D.: floor construction and field measurements; H.Z.: conceptualization, supervision, funding acquisition, review, and editing. All authors have read and agreed to the published version of the manuscript.

**Funding:** This study was funded by Inner Mongolia S&T Project (2022YFDZ0002) and National Natural Science Foundation (31770603).

**Institutional Review Board Statement:** Not applicable.

**Informed Consent Statement:** Not applicable.

**Data Availability Statement:** Not applicable.

**Acknowledgments:** The testing facility and material in this research were supported by the National Engineering Research Center of Wood Industry. These contributions are gratefully acknowledged. For the experimental study, the help of the following individuals is acknowledged: Du Yonghe, Wang Zhiyong, Han Xu, Wang Jiahui, Wang Duoyu.

**Conflicts of Interest:** The authors declare no conflict of interest.

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
