# Peer review of "Experimental Study and Numerical Simulation Analysis on Vertical Vibration Performances of 12 m Span Wood Truss Joist Floors"

_buildings, doi:10.3390/buildings12091455_

Round 1

Reviewer 1 Report

Comments

1. There is a large error in the experiment and analysis results, so it is necessary to correct it.

2. It is necessary to compare the analysis model and experimental results for all proposed models.

3. Since the level of acceleration of the proposed floor plate is very large, it is necessary to evaluate serviceability based on the acceleration acceptance criteria.

4. At the time of walking vibration, T1-T3 has the shape of a resonance response, whereas T4-T5 has the shape of an impulse response. Detailed analysis and comparison of these results are required.

5. Acceleration unit shall be m/s2 in Figures 16-18

Author Response

Response to Reviewer 1 Comments

Point 1: There is a large error in the experiment and analysis results, so it is necessary to correct it.

Response 1: We mainly show the vibration performances of the large-span wood truss joist floor. It is regretted to say that the numerical simulation technique in my teammate is still immature. We will refine the model in follow-up research.

Point 2: It is necessary to compare the analysis model and experimental results for all proposed models.

Response 2: We compared the analysis model and experimental results in modal results and 1 kN static load results. Is there anything else to add?

Point 3: Since the level of acceleration of the proposed floor plate is very large, it is necessary to evaluate serviceability based on the acceleration acceptance criteria.

Response 3: We added the evaluation of serviceability based on the acceleration acceptance criteria in 3.3.

Point 4: At the time of walking vibration, T1-T3 has the shape of a resonance response, whereas T4-T5 has the shape of an impulse response. Detailed analysis and comparison of these results are required.

Response 4: We increase some discuss about the difference in 3.3.

Point 5: Acceleration unit shall be m/s2 in Figures 16-18

Response 5: Acceleration unit in Figures 16-20 has been fixed.

Reviewer 2 Report

Dear authors,
In the beginning, I would like to congratulate you on the very interesting paper with a comprehensive approach allowing you to see not only the analytical and numerical solutions to the problem but also very professionally performed experimental tests. Below I would like to point out some elements worth improving.

A)    General remarks
1.    The main question that must be answered is how the authors' approach is different from typical case study tests. The reviewer is not convinced about the novelty of the study. In the last paragraph of the introduction, the authors, are providing the aim and scope of the study but do not present in a strong manner what is the novelty of the presented case. Please make sure this is presented both here and later in the conclusions.
2.    Article is clearly written and easy to follow. The authors give relevant references which are linked to their study, however, the quantity of references and cross-checking of the thesis made in the text is hardly sufficient. It must be pointed out that the majority of the references are from Chinese authors. It is suggested to look also on the worldwide literature. Some additional possibilities of references can be found in the following points but it is also suggested to add or change some references to present the achievements of other scientific centres.
3.     The abstract is well written introducing the basic overview of the paper. It is also written in a way that even a person not familiar with the topic can understand what the authors are proposing in their research. However, no significant novelty if the study is stated in the abstract.
4. The authors are asked not to use personal pronouns in the scientific text, this is not correct. Eg. Line 66 pronoun “we” occurs. Please recheck the whole article.
5.    The introduction provides basic background and overview of the methods used by the authors. However, the introduction is not presenting the state of the art for some measurement techniques in the case of modal analysis. Look at point 7.
6. Chapter  2 is clear. However, the title of this chapter is confusing when reading the article for the first time. The “Test floors” phrase is suggested to be changed to experimental setup or something that is more general.
7.    The research design is appropriate with the methodology explained and presented. However, there is no significant introduction to measurement techniques not in a specific chapter and especially not in the introduction. No presentation of alternative methods or measurement techniques since the authors are using a classical approach (modal hammer/accelerometers). Eg, modern, contactless methods should be evaluated for state of art analysis in the introduction. Especially the use of 3D Laser Vibrometry (perfect for free-free conditions where the sample can be supported on elastic strings) and Digital Image Correlation (DIC). You can use example for measurements on composite truss M.Guinchard "Non-invasive measurements of ultra-lightweight composite materials using Laser Doppler Vibrometry system" Proceedings of the 26th International Congress on Sound and Vibration (ICSV19) and modal analysis on steel blades "Quality Assurance and Control of Steel Blade Production Using Full Non-Contact Frequency Response Analysis and 3D Laser Doppler Scanning Vibrometry System" 2021 11th IEEE International Conference on Intelligent Data Acquisition and Advanced Computing Systems. The use of 3D laser vibrometry is especially beneficial for users who want to evaluate and directly connect measurements and simulations. In the case of DIC, you can use Emilio Di Lorenzo et. Al, “Full-Field Modal Analysis by Using Digital Image Correlation Technique”
8.    The simulation methods are described in detail. 
9.    There are no significant remarks to the results which are clearly presented. However, in fig 15, no legend allowing to see the level of amplitude is supplied. Is it possible to see extremal values on the picture so the mode shape is clearly visible? Maybe from a different angle.The same with the fig.22.
10.     In the case of the conclusions the authors emphasize what was done in the paper and the result presentation. It is more discussion of the results than real conclusions. The conclusions should also emphasize the usefulness of the results and their application. This should also inform the reader on what was the novelty of the study and are possible next steps. If the authors decide to leave the conclusions as they are I would suggest adding a few sentences on this matter.

B)    Item remarks
Fig.2 it is suggested to enlarge the drawings. Currently, the font size is too big with small drawings. It looks not fully professional. Also, both presented elements have different values sizes. Please, unify those two and also unify with the following fig3. And 4 so all presented figures have the same font, font size etc.
Fig.9 please correct the figure so it is of better quality and clearly presents the measurement points. The same with fig.11. Again please unify the style with other figures. Look at Fig.9 and 11 they should look similar but are prepared in a completely different style.
Fig16-20 again please increase the quality. The values and axis descriptions are not visible.

C)    Conclusions
The article is clear and interesting with no significant errors found in the research. Both methodology and results acquisition is correct. However, some changes have to be made in the case to the explanation of the novelty of the study. Moreover, some additional state-of-the-art analyses of modern modal analysis testing techniques and other used methods have to be incorporated in the introduction. Also, many of the figures require some additional work for proper, professional presentation of the results. At the current stage, the reviewer asks for major changes in those areas and will be happy to accept the paper after sufficient corrections.

Author Response

Response to Reviewer 2 Comments

Point 1: The main question that must be answered is how the authors' approach is different from typical case study tests. The reviewer is not convinced about the novelty of the study. In the last paragraph of the introduction, the authors, are providing the aim and scope of the study but do not present in a strong manner what is the novelty of the presented case. Please make sure this is presented both here and later in the conclusions.

Response 1: We emphrase the rare of large-span wood truss joist floor field test at the last paragraph of the introduction. In the conclusions, We edit the first paragraph and increase a new paragraph at last to callback the importantce of large-scale field test.

Point 2: Article is clearly written and easy to follow. The authors give relevant references which are linked to their study, however, the quantity of references and cross-checking of the thesis made in the text is hardly sufficient. It must be pointed out that the majority of the references are from Chinese authors. It is suggested to look also on the worldwide literature. Some additional possibilities of references can be found in the following points but it is also suggested to add or change some references to present the achievements of other scientific centres.

Response 2: We increase some recommended references to expand the research status.

Point 3: The abstract is well written introducing the basic overview of the paper. It is also written in a way that even a person not familiar with the topic can understand what the authors are proposing in their research. However, no significant novelty if the study is stated in the abstract.

Response 3: The journal encourage authors to use the structured abstracts to introduce background, methods, results and conclusions, so the cost is the loss of novelty.

Point 4: The authors are asked not to use personal pronouns in the scientific text, this is not correct. Eg. Line 66 pronoun “we” occurs. Please recheck the whole article.

Response 4: Personal pronouns in the whole article have deleted.

Point 5: The introduction provides basic background and overview of the methods used by the authors. However, the introduction is not presenting the state of the art for some measurement techniques in the case of modal analysis. Look at point 7.

Response 5: We added measurement techniques in the case of modal analysis in the introduce part.

Point 6: Chapter 2 is clear. However, the title of this chapter is confusing when reading the article for the first time. The “Test floors” phrase is suggested to be changed to experimental setup or something that is more general.

Response 6: The title of chapter 2 has changed to “Overview of the wood floor and test methods”

Point 7: The research design is appropriate with the methodology explained and presented. However, there is no significant introduction to measurement techniques not in a specific chapter and especially not in the introduction. No presentation of alternative methods or measurement techniques since the authors are using a classical approach (modal hammer/accelerometers). Eg, modern, contactless methods should be evaluated for state of art analysis in the introduction. Especially the use of 3D Laser Vibrometry (perfect for free-free conditions where the sample can be supported on elastic strings) and Digital Image Correlation (DIC). You can use example for measurements on composite truss M.Guinchard "Non-invasive measurements of ultra-lightweight composite materials using Laser Doppler Vibrometry system" Proceedings of the 26th International Congress on Sound and Vibration (ICSV19) and modal analysis on steel blades "Quality Assurance and Control of Steel Blade Production Using Full Non-Contact Frequency Response Analysis and 3D Laser Doppler Scanning Vibrometry System" 2021 11th IEEE International Conference on Intelligent Data Acquisition and Advanced Computing Systems. The use of 3D laser vibrometry is especially beneficial for users who want to evaluate and directly connect measurements and simulations. In the case of DIC, you can use Emilio Di Lorenzo et. Al, “Full-Field Modal Analysis by Using Digital Image Correlation Technique”

Response 7: Modern contactless methods of modal test are novel, we add this part in the introduction part and look forward to its application in wood floor modal tests. We are not cite Lukasz Scislo’s article because we can not get the full test.

Point 8: There are no significant remarks to the results which are clearly presented. However, in fig 15, no legend allowing to see the level of amplitude is supplied. Is it possible to see extremal values on the picture so the mode shape is clearly visible? Maybe from a different angle.The same with the fig.22.

Response 8: Due to the scaling of Mode Shapes, the displacement in mode shape just represents a proportional relation. The mode shapes of test floors we tested put out from DASP software are not given the level of amplitude. How the shape looks like is what we are concerned about, not the value.

Point 9: In the case of the conclusions the authors emphasize what was done in the paper and the result presentation. It is more discussion of the results than real conclusions. The conclusions should also emphasize the usefulness of the results and their application. This should also inform the reader on what was the novelty of the study and are possible next steps. If the authors decide to leave the conclusions as they are I would suggest adding a few sentences on this matter.

Response 9: We have added an extra paragragh at the last of conclusions to emphasize the usefulness and possible next steps.

Point 10: Fig.2 it is suggested to enlarge the drawings. Currently, the font size is too big with small drawings. It looks not fully professional. Also, both presented elements have different values sizes. Please, unify those two and also unify with the following fig3. And 4 so all presented figures have the same font, font size etc.
Fig.9 please correct the figure so it is of better quality and clearly presents the measurement points. The same with fig.11. Again please unify the style with other figures. Look at Fig.9 and 11 they should look similar but are prepared in a completely different style.
Fig16-20 again please increase the quality. The values and axis descriptions are not visible.

Response 10: The style of Fig.2-4 has unified. We emphrase the measure point in Fig.9 and Fig.11. The font size in Fig.16-20 was increse.

Round 2

Reviewer 2 Report

Dear Authors,

Thank you for revising the previous version of the manuscript.

Concerning the previous comments, all the points were addressed and the reviewer can state that the paper is of sufficient quality for publication in the present form.

Best regards,

The reviewer